# A Systematic Review of Literature on the Association Among Sleep, Cortisol Level and Cardiovascular Health Within the Healthcare Shift Worker Population

**DOI:** 10.3390/biomedicines13102539

**Published:** 2025-10-17

**Authors:** Aslah Nabilah Abdull Sukor, Norsham Juliana, Nazefah Abdul Hamid, Nur Islami Mohd Fahmi Teng, Muslimah Ithnin, Sahar Azmani, Sazzli Shahlan Kasim

**Affiliations:** 1Department of Physiology, Faculty of Medicine, Universiti Teknologi MARA, Sungai Buloh Campus, Jalan Hospital, Sungai Buloh 47000, Malaysia; aslahnabilah@uitm.edu.my; 2Faculty of Medicine and Health Sciences, Universiti Sains Islam Malaysia, Persiaran Ilmu, Putra Nilai, Nilai 71800, Malaysia; nazefah@usim.edu.my (N.A.H.); drazmanisahar@usim.edu.my (S.A.); 3Department of Nutrition & Dietetics, Faculty of Health Sciences, Universiti Teknologi MARA, Puncak Alam 42300, Malaysia; nurislami@uitm.edu.my; 4School of Health Sciences, KPJ Healthcare University, Persiaran Seriemas, Kota Seriemas, Nilai 71800, Malaysia; drmuslimah@kpju.edu.my; 5Department Cardiovascular and Thoracic, Universiti Teknologi MARA, Puncak Alam 42300, Malaysia; sazzlishahlan@uitm.edu.my

**Keywords:** sleep quality, insomnia, shift, cortisol, cardiovascular, circadian rhythm

## Abstract

Shift workers are commonly associated with circadian misalignment due to irregular working hours, which often leads to poor sleep quality and is associated with HPA axis misalignment and changes in cardiovascular outcome. **Background:** This systematic review aimed to investigate the association between cortisol production and cardiovascular health with sleep quality in healthcare shift workers. **Methods:** A comprehensive search of PubMed, Web of Science, and Scopus was conducted for studies published between 2010 and 2025, according to PRISMA guidelines. Fourteen studies met the inclusion criteria. **Results:** Among the included studies, eight studies focused on the relationship between sleep quality and cortisol regulation, five studies investigated the link between sleep quality and cardiovascular health, and one study examined sleep quality, cortisol regulation, and cardiovascular outcome. A significant relationship between cortisol and sleep quality was observed, as lower cortisol levels upon awakening were associated with low sleep quality. Several studies reported that sleep disturbances were associated with adverse cardiovascular outcomes, including reduced heart rate variability (HRV) and increased risk of metabolic syndrome. **Conclusions:** This review highlights existing literature on the critical role of sleep quality as a key factor in cortisol level and cardiovascular health in shift workers, along with the factors influencing circadian rhythm.

## 1. Introduction

A shift work schedule is defined as working hours beyond the conventional daytime hours of 6 a.m. to 6 p.m., and it involves a rotating schedule [1]. It has long been part of the global workforce operation, with approximately 30% of workers engaged in this type of schedule [2,3]. Despite the disadvantages to health, such as disruption of daily circadian rhythm, shifting working hours around the clock is imperative to meet the continuous demand of the urban world and crucial services [1,4]. Circadian rhythm is defined as an internal biological rhythm that regulates a 24 h period of sleep–wake cycle controlled by an endogenous clock, which affects hormones and metabolism [5,6].

Healthcare workers engaged in rotating-shift schedules often experience disrupted routines and irregular sleep patterns [7]. Their roles are typically accompanied by a high workload and require continuous focus, which may impact both physical and mental well-being. Part of the health consequences of disruptive circadian rhythms include dysregulation of cortisol [8] and increased cardiovascular risk [9].

Cortisol is the primary hormone associated with wakefulness and is closely linked with an individual’s sleep cycle [10]. It regulates cortisol secretions, with the highest during the day and lowest at night. However, recent findings by Li J et al. [11] reveal that working in shifts alters the diurnal pattern of cortisol. Moreover, 72% of shift workers reported suffering from insomnia and insufficient sleep [12]. Shift workers are prone to tiredness due to inadequate sleep quality and circadian rhythm disruption. Night shift workers often require three to four days to adjust their cortisol circadian rhythm [13]. Nurses working night shifts displayed elevated morning cortisol levels compared to those on standard hours [11]. The importance of cortisol as a potential marker of circadian dysregulation in shift workers was highlighted when Niu et al. [14] showed that cortisol secretion in night shift workers has a lowered peak in the morning and an elevated peak at night compared to permanent shift workers. Fixed shift workers are defined as workers who work the same shift schedule without rotation [15]. However, one of the factors that generates changes in the diurnal cortisol pattern is shift work [11]. Shift work is known to affect sleep quality and cause circadian disruption. Circadian disruption is the misalignment of both internal disruption (hormone, metabolic, and sleep/wake cycle) and external behaviours (schedule, feeding, and light exposure) of biological timing with altered rhythm amplitude [16].

Sleep–wake rhythms and sleep patterns play an important role in maintaining cardiovascular health. Good quality sleep helps in regulating normal blood pressure, heart rate, metabolism, and inflammatory pathways [17]. Chronic sleep disturbances can cause HPA axis misalignment, resulting in cortisol dysregulation that can exacerbate cardiovascular disease [18]. These physiological changes that are initiated by poor sleep and continuous circadian misalignment increase risk for hypertension, atherosclerosis, and coronary heart disease over time [19]. Furthermore, poor sleep quality, which is characterized by increased wake after sleep onset (WASO), low sleep efficiency, and non-restorative sleep, is a potential cardiovascular risk [20]. Another contributing factor, such as high workload, fatigue, and short period of recovery, may amplify the effects and elevate cardiovascular burden among this population [21].

Therefore, the objective of this systematic review is to investigate the association between sleep quality, cortisol regulation, and cardiovascular outcomes. In addition, to gain a deeper understanding of circadian disruption among shift workers, this study aims to identify gaps in the existing literature and provide evidence-based recommendations for workplace interventions and future research directions.

## 2. Methods

### 2.1. Study Selection Criteria

The inclusion criteria for the studies were (1) full-time healthcare shift workers (including those with irregular rotating shifts and day, evening, and night hours); (2) quantitative research articles focused on healthcare shift workers that explored peripheral biomarkers related to the sleep–wake cycle; (1a) biomarkers were measured in hair, saliva, urine, blood, serum, or plasma; (1b) the study included parameters related to sleep or the sleep–wake cycle using the Athens Insomnia Scale (AIS), Pittsburgh Sleep Quality Index (PSQI), or Epworth Sleepiness Scale (ESS); (1c) cardiovascular parameters; and (2) publications in English. The exclusion criteria were (1) review, meta-analysis, letters, newsletters, editorials, conference abstracts, or case studies papers (2) animal studies and (3) intervention studies for sleep effect; (4) a qualitative or mixed-methods study, as the focus of this review was on quantitative studies providing objective biomarker data (e.g., cortisol, cardiovascular outcomes) that allowed for systematic comparison across studies. While qualitative studies can provide valuable psychosocial and contextual insights into shift work, they exceed the biomarker-focused scope of this review.

### 2.2. Search Strategy

The International Prospective Register of Systematic Reviews (PROSPERO) has registered this systematic review under the registration number CRD42024561659. The review was explored exclusively through three electronic databases: PubMed, Scopus, and Web of Science included articles published from 2010 to 2025. The search query involves terms including three sets: (1) “sleep disorder,” “sleep disturbance,” “sleep hygiene,” and “sleep quality”; (2) “cortisol,” “corticoid,” and “corticosteroid”; (3) “cardiovascular,” “heart,” and “cardiac”; (4) “shift work,” “night work,” “rotating shift work,” and “night shifts.” This query aimed at identifying the relationship between (i) sleep quality and cortisol and (ii) sleep quality and cardiovascular health among healthcare shift workers.

### 2.3. Study Selection Process

The electronic database search from 2010 to 2025 generated 688 potentially relevant articles from the keywords. In the first phase, 20 articles were excluded, as the articles were duplicated in [22] and the abstracts of the other 668 were screened. Upon screening the abstract, 613 papers were excluded, as the articles were not related to the studies. The full text of the other 55 papers was then retrieved. After a thorough assessment of the full text, 41 articles that did not meet the criteria were excluded for several reasons: articles that are not original research, articles not written in English, and articles that did not follow the study design. Finally, fourteen studies were selected for this systematic review. The steps involved in the article selection process are shown in Figure 1, which shows a summary of the characteristics of the included studies. A total of 14 relevant journal articles were included in the review (Table 1, Table 2 and Table 3).

### 2.4. Data Extraction Process for Each Study

Studies fulfilling the inclusion criteria were extracted. All authors carried out data extraction. The process was performed independently, and a data collection form was used. Discussions between reviewers took place to resolve any disagreements. The following data was extracted from the selected articles: (1) Authors and year of publication, (2) the sample size, (3) exposure to shift work, (4) the measurement of cortisol, (5) the measurement of the cardiovascular parameter, (6) the measurement of the sleep parameter, (7) the summary of the outcome, (8) conclusion.

### 2.5. Study Quality

Each paper included in this review was evaluated for study quality by Aslah Nabilah Abdull Sukor (ANAS) and Muslimah Ithnin (MI) using the Joanna Briggs Institute (JBI) critical appraisal tools [36]. The quality outcomes were then validated by Norsham Juliana (NSJ) and Nazefah Abdul Hamid (NAH). Overall, low quality (high risk of bias) was considered less than 50%, 50–69% was rated as a moderate quality (moderate risk of bias), and lastly, more than 69% was rated as a high-quality (low risk of bias) paper. The detailed scoring for each individual study is available in Appendix A.

## 3. Results

### 3.1. Studies Characteristics

Study subjects included healthcare shift workers, e.g, nurses, physicians, medical technicians, and lab workers. The majority of studies focus on nurses (12 out of 14 studies). The studies conducted in Taiwan [16,29,34,37], Italy [24,25], United Kingdom [30], United Arab Emirates [27], India [33,35] Croatia [26], China [37], Brazil [25], and Canada [31]. The primary outcome for all papers in this review was sleep quality, cortisol and cardiovascular outcome. Cortisol levels were evaluated using saliva (*n* = 7) and blood (*n* = 2; plasma or serum unspecified). The studies varied in their measurement methodologies, including the biological matrix used (saliva vs. blood), the timing of sample collection (morning, post-shift, or throughout schedules), and the sampling frequency (single vs. multiple time points). Tools applied to measured sleep quality included the Pittsburgh Sleep Quality Index (PSQI), Athens Insomnia Scale (AIS), Epworth Sleepiness Scale (ESS), Shiftwork Sleep Index (SSI), and actigraphy. For saliva cortisol sampling, three studies employed two sampling time points [23,24,29,30] and another two studies collected saliva samples at four time points [25,37]. Three studies collected samples on awakening [29,30,37]. The cardiovascular measurement used heart rate variability, metabolic indicators, self-reported data, and medically reported data. The results of the studies will be presented by outcome: (1) cortisol levels in the morning and in the evening; (2) relationship between cortisol and sleep quality; (3) relationship between cortisol and shift schedule; (4) relationship between cardiovascular and sleep quality; (5) relationship between shift schedule and sleep.

### 3.2. Cortisol Level in the Morning and in the Evening in Healthcare Shift Workers

From the fourteen studies reviewed, six studies measured the cortisol in the morning and in the evening. In the studies exploring the morning cortisol levels, the five studies that analysed cortisol levels had different findings. Vivarelli et al. [24] reported that 17% of the nurses had morning cortisol above the threshold limits, with a mean morning cortisol of 0.492 µg/dL (range: 0.001–2.240 µg/dL); however, they did not specify the exact numerical threshold used to define ‘above cutoff’. However, Lowson et al. [30] discovered that the early morning cortisol level following night work was 10.7 ± 5.7 nmol/L, significantly lower (*p* < 0.05) than that of subsequent mornings. It should be noted that the early morning saliva cortisol was obtained later than on other mornings, specifically at 08:16 ± 34 min following night shifts, as opposed to 07:41 ± 34 min for other mornings.

Bani-Issa et al. [27] reported that 36.1% had impaired (below average) levels of morning cortisol, which were taken between 7 a.m. and 8 a.m. According to a study by Tsai et al. [37], the morning cortisol level at 0 min after waking was very low (*p* < 0.05). Chang et al. [29] also found morning cortisol levels upon waking and 30 min after waking were low.

Regarding the evening cortisol levels, all three studies reported high levels. Saptarshi Roy et al. [35] reported a high evening cortisol level (sig = 0.036), and Lowson et al. [30] reported higher late evening cortisol (4.9 ± 7.2 nmol/L) compared to other times (3.0 ± 1.5 nmol/L) but not significant (*p* = 0.102). However, those samples that were collected earlier during night works (21:21 ± 1 h 32 min) were significant (*p* < 0.05) compared with other times (22:22 ± 44 min). Bani-Issa et al. [27] found 14.3% of shift workers had high bedtime cortisol around 7 p.m. to 8 p.m.

### 3.3. Relationship Between Cortisol Level and Shift Schedule

Of the fourteen studies reviewed, four papers investigated the relationship between cortisol level and shift schedule. Shift type influenced cortisol level. In the study by Ljevak et al. [26], blood cortisol concentrations were measured in shift workers who worked 12-h night shifts. Moreover, Zhang et al. [23] observed that 12 h shift workers exhibited elevated morning saliva cortisol levels before the day shift (median 0.54 vs. 0.31, *p* < 0.005) and increased evening saliva cortisol levels following the night shift (median 0.51 vs. 0.31, *p* < 0.005) in comparison to 8 h shift nurses. Interestingly, Chang et al. [29] had studied three different shifts, which were day, evening, and night shifts. The cortisol levels of day shift workers 30 min post-waking were significantly elevated compared to those of evening and night shift workers, while the cortisol levels of night and evening shift workers were significantly lower than those of day shift nurses. The study by Minelli et al. [25] found that cortisol production during the night shift has a positive correlation with baseline diurnal cortisol secretion measured on non-working days, suggesting that an individual’s baseline cortisol rhythms may influence the hormonal response during night shift work.

### 3.4. Relationship Between Cortisol and Sleep Quality

Of the fourteen papers reviewed, two papers studied the relationship between cortisol and sleep quality. The result showed that 54.3% of subjects with low sleep quality, defined by shorter total sleep time (TST) and longer sleep onset latency (SOL) with higher cortisol levels at awakening (9.15 + 3.33 ng/mL), exhibit a flatter cortisol awakening response (CAR) and a smaller diurnal slope 30 min to 12 h than subjects who have high sleep quality, defined by longer TST and shorter SOL [37]. The high sleep quality group has a cortisol concentration higher at 30 min after awakening (11.79 + 5.80 ng/mL) and lower at 12 h after awakening (2.40 + 2.48 ng/mL) than the low sleep quality group. Chang et al. [29] found cortisol level significantly correlated (ΔF = 5.17, *p* = 0.025) with sleep quality. Individuals with higher morning cortisol levels had better sleep quality, and those evening or night shift workers who displayed significantly lower morning cortisol levels had significantly poorer sleep quality than day shift workers. Day shift workers are those who work regular daytime hours, typically from morning to late afternoon.

### 3.5. Relationship Between Cardiovascular Health and Sleep Quality

The collective findings from six studies highlight a relationship between sleep health and cardiovascular health. Poor sleep quality encountered by night shift workers is associated with disrupted heart rate variability (HRV). Roy et al. [35], Panwar et al. [33] and Hsu et al. [16] reported that night shift nurses presented lower parasympathetic activity and higher sympathetic tone characterized by decreased SDNN and HF power and increased LF/HF ratio. Although interventions like low-level LED light therapy help in improving sleep, they did not produce significant changes in the HRV [34].

A study by Silva Costa et al. [32] reported both insomnia and night work were independently associated with higher prevalence of CVD. There is a 66% increased risk of cardiovascular disease among nurses who had insomnia, even after adjusting for their age, BMI, lifestyle factors, and work schedule. According to the study, insomnia is not just a symptom experienced at night but is also an independent risk factor for cardiovascular health. This is similar to the study by Lajoie et al. [31] that linked insomnia and poor sleep quality with increased cardiometabolic risk. The study found that sleep quality was a stronger predictor of cardiometabolic risk.

### 3.6. Shift and Sleep

All fourteen papers studied the relationship between shift work and sleep. Roy et al. [35] found individuals working shifts (two morning: 6 a.m.–2 p.m., two evening: 2 p.m.–10 p.m., and two-night duties: 10 p.m.–6 a.m.) per week displayed a significantly (*p* < 0.001) higher chance of having insomnia (Athen’s score > 6) than fixed-duty workers (9 a.m.–5 p.m.). One study found that insomnia was common and more severe among those who are exposed to long-term night shifts, with 35% having diurnal insomnia and 24% having nocturnal insomnia among night shift workers [32]. Studies by Chang et al. [29], Panwar et al. [33], Hsu et al. [16] and Lajoie et al. [31] reported night shift workers displayed significantly poorer sleep quality than day or evening shift workers in terms of sleep duration, daytime dysfunction, and overall sleep quality. Zhang et al. [23] found individuals working an 8 h shift had significantly more total sleep time (TST) (456 vs. 364 min) but a longer reaction time before the night shift compared to individuals working a 12 h shift. Another study found those who were working in shifts (12 working hours to 24 h off-work, 12 working night shift hours to 48 h off-work) had higher sleep difficulties than those working a first shift (standard day working hours, 7:30 a.m. to 2:30 p.m.). They also found a strong correlation in the rate of sleep disturbances between individuals working in rotating shift schedules and morning shift workers [26]. In contrast, one study found 75% of night shift workers had good outcome scores on the PSQI, 21.6% had poor outcomes, and 2.9% had bad outcomes; however, the ESS reported that 85.3% experienced no daytime sleepiness [24]. Interestingly, a randomized controlled study reported that low-level LED light therapy had significantly improved the sleep quality [34].

## 4. Discussion

This systematic review explores the relationship between sleep health, cortisol levels, and cardiovascular health among healthcare shift workers. The results consistently indicate that shift work-induced circadian disruption leads to sleep problems, which are linked to dysregulated cortisol secretion and adverse cardiovascular outcomes. In this review, circadian disruption refers to misalignment between the body’s internal biological rhythms and externally imposed sleep–wake cycles, which leads to sleep problems, which are linked to dysregulated cortisol secretion and adverse cardiovascular outcomes [18,28]. This misalignment has been shown in multiple studies to impair physiological processes including glucose tolerance, hormonal regulation (including cortisol secretion), and cardiovascular risk [10,38,39]. Nevertheless, it is necessary to acknowledge that the current evidence is primarily correlational, and causal connections cannot be clearly determined from the included studies.

### 4.1. Sleep Health and Cortisol Level

Normal cortisol secretion follows a negative diurnal slope. In adults, normal conditions of cortisol pattern usually reach a peak around 8 a.m. to 9 a.m. (4.7–7.0 ng/mL) with an average of 5.24 ng/mL and begin to decline around noon, <4 ng/mL. However, abnormal cortisol levels were observed in shift workers. This review identified elevated evening or bedtime cortisol levels in shift workers [29,30,35] as well as low morning cortisol levels among night shift workers [27,30,35,37]. These patterns, characterized by low morning and high evening cortisol levels, result in a flatter diurnal slope, which is indicative of HPA axis dysregulation [40]. Such dysregulation may increase the risk of diabetes [41], coronary disease [42] and depression [43].

Evidence suggests a bidirectional interaction between sleep quality and cortisol levels. Chang et al. [29] reported cortisol levels significantly correlated with sleep quality, while Jui Tsai et al. [37] found that individuals with low sleep exhibited a flatter/blunted CAR, a flattened diurnal slope, shorter total sleep time (TST), and a longer sleep onset latency (SOL). The impaired cortisol patterns may contribute to sleep difficulties or sleep disturbances and increase the risk of insomnia, sleep deprivation, or sleep debt. Conversely, poor sleep quality may elevate cortisol response to stress, alter diurnal cortisol output, and reduce morning cortisol levels. The HPA axis stimulates cortisol secretion, promoting wakefulness, increasing brain activity, and reducing slow-wave sleep [44,45]. This cycle suggests that cortisol is both a mediator and an outcome of sleep disturbances in shift workers.

Shift workers on 12 h shifts, with 24 or 48 h recovery periods, often exhibit morning cortisol levels that deviate from the reference values [26]. Ljevak et al. [26] also found a strong correlation between rotating shift systems and sleep disturbances. These findings underscore the importance of shift timing in providing adequate recovery periods [46]. Understanding how the shift timing and duration affect sleep quality could help establish optimal shift durations. Notably, cortisol levels are significantly higher before day shifts and after night shifts in 12 h shifts compared to 8 h shifts [23]. While the underlying mechanism is not fully understood, hypotheses point to the complex endocrine interactions involving prolonged cortisol hypersecretion, which is affected by extended response to stress and higher fatigue levels. Most shift workers sleep less than the recommended 8 h, leading to circadian disruption, which may reduce work performance [47].

Minelli et al. [25] found that cortisol output during night shifts positively correlated with basal diurnal cortisol secretion on non-working days. This suggests a carryover effect of cortisol regulation from night shifts to subsequent days. Evidence indicates that adjusting working hours and rest times for shift workers can reduce sleepiness levels [48]. To realign the circadian rhythm of cortisol production, consecutive night shift workers require at least four days off to recover [11]. Studies indicate that complete circadian adjustment typically takes approximately 7 days, especially among shift workers transitioning from day to night shifts. The review by Folkard S. [49] noted that those permanent or long-term night shift workers rarely achieve full circadian adaptation; only 3% succeeded. This highlights the importance of individual factors such as the ability to maintain consistent night work schedules, day sleep during off days, and personal diurnal preferences in managing circadian disruption.

### 4.2. Sleep Health and Cardiovascular Health

Sleep quality is an important marker of cardiovascular health, especially among healthcare workers who have circadian disruption. Poor sleep quality characterized by reduced total sleep time, poor sleep latency, and low sleep efficiency was found to be associated with reduced heart rate variability, which is an indicator of autonomic imbalance. These physiological changes are significant indicators of increased cardiovascular risk.

Heart rate variability is a non-invasive biomarker of cardiovascular regulation and the autonomic nervous system. Studies among healthcare workers with poor sleep quality, including those by Panwar et al. [33] and Hsu et al. [16] reported significantly lower HRV values and elevated LF/HF ratios. These HRV parameter alterations reflect increased sympathetic activity and reduced parasympathetic activity linked to hypertension, myocardial infarction, stroke, and sudden cardiac death [50,51].

Poor sleep quality is also related to cardiometabolic dysregulation. For example, Lajoie et al. [31] discovered that female hospital workers exposed to shift work with poor sleep quality had increased waist circumference and reduced HDL cholesterol levels. These findings are consistent with large cohort studies that associate insomnia with elevated metabolic syndrome and coronary artery disease [52,53]. The mechanism includes increased systemic inflammation (C-reactive protein, IL-6, and TNF-α), hormone imbalance, and endothelial dysfunction [54].

Interestingly, Liao et al. [34] showed that although light therapy improved sleep quality and psychological symptoms among shift-work nurses, it did not produce significant changes in HRV, suggesting that short-term improvements in sleep may not be sufficient to reverse long-standing autonomic dysfunction. This aligns with findings by Morris et al. [55], which emphasized the importance of sustained interventions to observe significant improvements in cardiovascular biomarkers among shift workers.

### 4.3. The Sleep, Cortisol and Cardiovascular Feedback Loop

The study presented in this review underlines a cyclical feedback cycle that links shift work, circadian misalignment, cortisol dysregulation, poor sleep quality, and adverse cardiovascular outcomes. The hypothalamic–pituitary–adrenal (HPA) axis is disrupted because of shift work, which disrupts the circadian rhythm. This disruption results in abnormal cortisol secretion patterns, with elevated evening levels and lowered morning peaks, which form a reinforcing cycle. These cortisol imbalances result in impaired sleep quality by increasing sleep latency, decreasing sleep duration, and decreasing restorative slow-wave sleep [56]. A bidirectional relationship takes place when poor sleep quality triggers an increase in stress reactivity and further disrupts the cortisol rhythm [57]. Related to the HPA axis, cardiovascular health disorders can be caused by the combined effect of poor sleep and cortisol impairment, which result in increased sympathetic activity, reduced parasympathetic tone, heightened systemic inflammation, and endothelial dysfunction. These changes increase the risk of hypertension and metabolic disorder. This feedback loop is important for hospital shift workers, where recovery time limits and physiological demands are high [58].

To illustrate the integrative findings, Figure 2 presents a conceptual model showing a cyclical feedback loop. This self-reinforcing cycle highlights the complex interactions linking shift work to health risks in healthcare workers. The table summarizes the distribution of included studies reporting the following associations: (i) cortisol and sleep, (ii) sleep and cardiovascular outcomes, and (iii) the integrated relationship between sleep, cortisol, and cardiovascular diseases. Although this conceptual model provides a useful framework to integrate the observed associations, it is important to acknowledge that the evidence is primarily correlational. It is impossible to establish causal pathways that connect shift work, cortisol dysregulation, sleep, and cardiovascular health.

## 5. Limitation

There are several limitations in this review. First, the database search was restricted to PubMed, Scopus and Web of Science, which may have excluded relevant studies from other database sources. This may restrict the scope of retrieved literature. Additionally, the search strategy also should consider a broader range of synonyms and related concepts to increase comprehensiveness. Second, there was substantial heterogeneity presented across the studies, as cortisol was measured in different biological samples (saliva, blood) with varying timing and protocols, while sleep was measured using diverse tools (PSQI, ESS, AIS, actigraphy), and cardiovascular health also ranged widely from objective measures (HRV) to self-reported diagnoses. These inconsistencies complicate direct comparisons across studies and may contribute to variability in findings. Third, no formal publication bias was evaluated due to the absence of meta-analysis, which may increase the possibility that positive results are overrepresented. Fourth, most of the included studies lacked standardized statistical reporting (effect sizes and confidence intervals), limiting the ability to evaluate the robustness of associations. Fifth, the majority of the included studies concentrated on nurses, potentially limiting generality to other healthcare worker groups, yet the findings remain relevant considering the high proportion of nurses in shift work. Moreover, the review primarily focused on cortisol as a key biomarker, while other mediators (melatonin, inflammatory cytokines, and autonomic markers) were inadequately included. Finally, lifestyle confounders such as diet, caffeine, psychosocial stress, and physical activities, which are known to affect cortisol, sleep, and cardiovascular health, were unaccounted for, which may bias the reported outcome.

## 6. Conclusions

This systematic review assessed objective cortisol parameters (salivary and blood biomarkers) together with subjective psychodiagnostics measures (questionnaires) in healthcare shift workers. All studies included in the review reported that cortisol levels are affected to some extent, correlating with reduced sleep quality among the shift workers. Notably, cortisol has the potential to serve as a biomarker for assessing sleep quality in this population. Because cortisol levels closely reflect individual variations in circadian rhythm, the hormone may be a valuable marker for identifying and monitoring health risks associated with circadian disruption. More studies are needed to determine the optimal methods and timing for cortisol sample collection to effectively detect circadian rhythm disruptions among shift workers with varying schedules.

In addition to highlighting future research needs, this review underscores the importance of (1) conducting longitudinal or intervention studies that clarify causal pathway and measure the effectiveness of targeted strategies such as optimized schedule of shift and workplace intervention such as light therapy, sleep hygiene and pharmacological modulation; (2) broadening the database and expanding the search query to include a wider range of related terms; (3) expand recruitment and diversify populations by including allied health professionals, physicians and other healthcare shift workers; (4) standardize protocols for sleep assessment and biomarker sampling (timing and biological samples), expanding the biomarkers panel to provide a comprehensive evaluation of physiological mechanisms.

## Figures and Tables

**Figure 1 biomedicines-13-02539-f001:**
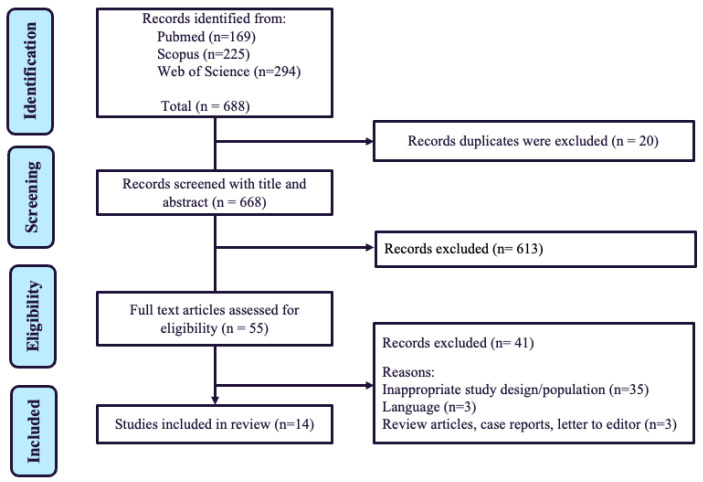
Flow chart of the study selection process based on PRISMA guidelines. (Abbreviations: *n* = number of studies).

**Figure 2 biomedicines-13-02539-f002:**
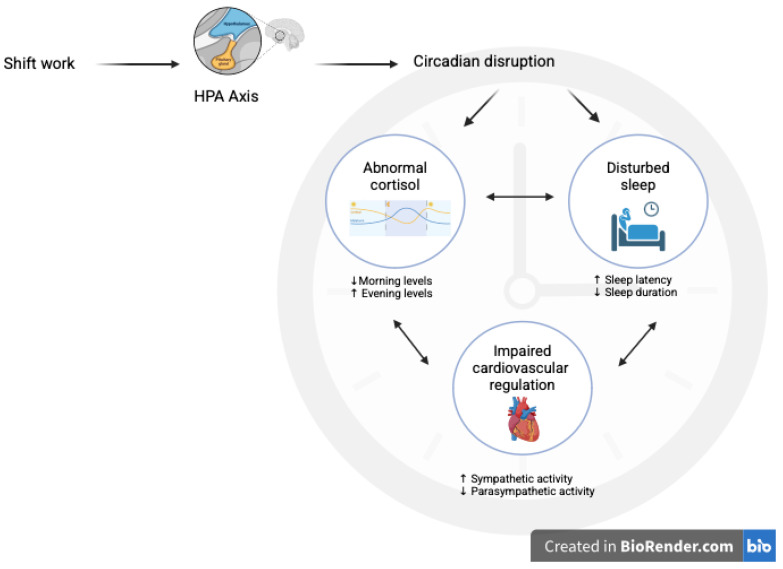
This cyclical feedback loop highlights how shift work disrupts biological rhythms, leading to sustained health risks. Created in BioRender. Cannizzaro E, et al. (2020) https://app.biorender.com/illustrations/63bc18e8ff5ed4fe57e7b69c.

**Table 1 biomedicines-13-02539-t001:** Summary of measurement of sleep quality and cortisol among healthcare shift workers of the included studies.

Author	Sample Size (*n*)	Exposure of Working Shift	Cortisol Measures	Sleep Quality Measures	Outcome	Conclusions
Zhang, 2023 (China) [23]	Nurses (*n* = 152)	Type of shift: 8 h (day–evening–night) and 12 h (day–night)	Saliva samples were collected between: 7:00–8:00 and 19:00–20:00	Sleep-wake indexes are analyzed by actigraphy data	Compared with the 8 h shift nurses, the 12 h shift nurses had: **Cortisol result:** Higher saliva cortisol levels before the day shift (median 0.54 vs. 0.31, *p* < 0.005) and after the night shift (median 0.51 vs. 0.31, *p* < 0.005). **Sleep result:** Significantly more TST (456 vs. 364 min) Longer reaction time before the night shift (286 vs. 277 min).	Fast rapid rotation in shift workers disturbed circadian rhythm and reduced total sleep time.
Vivarelli, 2023 (Italy) [24]	Nurses (*n* = 102)	Type of shift: Night shift with forward rotating shift. Morning shift followed by an afternoon shift and a night shift, then followed by a rest day	Saliva: morning and evening	PSQI ESS	**Cortisol result:** 17% had above the threshold limit of morning salivary cortisol. 91% had below the threshold limit of evening cortisol. **Sleep result:** PSQI: 75% good sleep, 21.6% poor sleep. 2.9% bad sleep. **ESS**: 85.3% no daytime sleepiness	There was a positive relationship between cortisol and sleepiness, even if not significant.
Minelli, 2021 (Italy) [25]	Nurses (*n* = 30)	Type of shift: Morning shift 06:00 Afternoon shift 14:00 Night shift 22:00	Saliva: Four salivary samples were collected at 21:00, 24:00, 03:00 and 06:00	PSQI	**Cortisol result:** Cortisol concentration is low around 21:00 and 24:00, then increasing during early morning between 03:00 and 06:00. Cortisol output during the early night shift failed to correlate with PSQI score. Cortisol output during night shifts positively correlates with basal cortisol diurnal secretion measured during non-working days. **Sleep result:** PSQI: 76% had good sleep.	Rotating shift workers showed that low total cortisol during night work is associated with circadian misalignment.
Ljevak, 2020 (Croatia) [26]	Subjects (*n* = 157): 135 female nurses. 22 male medical technicians	Control group: Morning shifts (7:30 to 14:30) Experimental group: 12 working hours to 24 h off-work and 12 working night shift hours to 48 h off-work	Blood: collected at 7.30 a.m.	Standard Shiftwork Index	**Cortisol result:** 5% of shift workers had morning cortisol levels lower than the reference values, and 5% had morning cortisol levels higher than the reference values. **Sleep result:** Shift workers showed significantly higher levels of sleep difficulties than the first-shift workers. A strong correlation was found in the rate of sleep disturbances between morning shift workers and those participants working a rotating shift system.	Shift workers had impaired cortisol levels and higher levels of sleep difficulties.
Bani-Issa, 2020 (UAE) [27]	Healthy adult women healthcare professionals (*n* = 335)	Day or night shifts	Saliva samples were collected at: (7:00–8:00) and (19:00–20:00)	PSQI	**Cortisol result:** Morning cortisol level: 36.1% had impaired below average levels. Bedtime cortisol level:14.3% had high bedtime cortisol levels. There was a significant correlation between morning and bedtime cortisol levels (r = 0.196, *p* = 0.001). **Sleep result:** 60.3% had poor sleep quality. Morning and bedtime cortisol levels were not significantly correlated with quality of sleep (r = 0.26, *p* = 0.57 for morning cortisol; r = 0.013, *p* = 0.92 for bedtime cortisol).	Shift workers had impaired morning and bedtime cortisol level with poor sleep quality. Cortisol levels in the morning and at night were independently correlated with sleep quality.
Tsai, 2019 (Taiwan) [28]	Female nurse (*n* = 61)	Worked 8 h per shift. 4 weeks of regular day-shift work (8:00–16:00)	4 saliva samples collected at: awakening and after awakening 0 min, 30 min, 6 h, 12 h.	PSQI Actigraphy data	**Cortisol result:** LSQ had a higher cortisol concentration at awakening (0 min) than did the HSQ. LSQ had a flatter CAR and smaller diurnal cortisol slope 30 min to 12 h than HSQ. HSQ had significantly higher cortisol concentration at 30 min after awakening (11.79 + 5.80 ng/mL) than LSQ. HSQ had significantly lower mean cortisol concentration at 12 h after awakening (2.40 + 2.48 ng/mL) than LSQ. **Sleep result:** 54.3% LSQ (CPSQI > 5), 45.7% HSQ (CPSQI 5), HSQ had greater TST, higher SE, and shorter WASO and SOL than the LSQ.	Low sleep quality showed flatter cortisol awakening response (CAR).
Chang, 2018 (Taiwan) [29]	Female nurses (*n* = 128)	Types of shifts: 08:00–16:00 16:00–24:00 00:00–08:00	Saliva: Collected after waking and 30 min after waking.	PSQI	**Cortisol result:** CARi night shift and the evening shift workers is significantly lower than the day shift. Shift type significantly influenced CARi (F = 19.66, *p* < 0.001). **Sleep result:** Evening shift or night shift workers have significantly poorer sleep quality than those working the day shift. Shift type significantly influenced sleep quality (F = 15.13, *p* < 0.001).	There was a significant correlation between CARi and sleep quality; nurses with higher CARi had better sleep quality.
Lowson, 2013 (UK) [30]	Nurses (*n* = 20)	Types of shifts: Nurses (*n* = 15) 7 to 8 h and night shifts of 10 to 12 h, 07.00 to 21.30 h and finishing between 07.00 to 08.00. Nurses (*n* = 5) 12 h shifts with shift changes 20:00 and 08:00	Saliva: Collected just after waking in the morning and just before sleeping in the evening.	PSQI	**Cortisol result:** Cortisol level soon after waking significantly lower (time 07:58 ± 38 min) (12.5 ± 5.6 nmol/L) (*p* < 0.001) than the late evening 21:53 ± 1 h 17 min) 4.0 ± 5.2 nmol/L During periods of night work: The early morning cortisol levels for nurses was 10.7 ± 5.7 nmol/L which was significantly lower (*p* < 0.05) than the early morning cortisol level (14.2 ± 5.0 nmol/L) for other times the early morning saliva samples were collected significantly later (*p* < 0.01) compared with other mornings (08:16 ± 34 min) following night shifts compared with 07:41 ± 34 min for other mornings). For late evening cortisol levels, higher cortisol levels (4.9 ± 7.2 nmol/L) were found during periods of night work (non-significant) (*p* = 0.102) compared with other times (3.0 ±1.5 nmol/L). **Sleep result:** During periods of night work, sleep quality worse (nonsignificant) and more sleepiness (Karolinska Sleepiness Scale) before (*p* < 0.01) and after (*p* < 0.001) their main period of day sleep compared with night-time sleep-in periods of no night work.	Night shift workers reported to have worse sleep quality (non-significant) and lower early morning cortisol levels than other mornings.

Abbreviations: cortisol levels (CARi); Pittsburgh Sleep Quality Index (PSQI); Epworth Sleepiness Scale (ESS); cortisol awakening response (CAR); low sleep quality (LSQ); high sleep quality (HSQ); Chinese version of the Pittsburgh Sleep Quality Index (CPSQI); total sleep time (TST); sleep efficiency (SE); wake after sleep onset (WASO); sleep onset latency (SOL).

**Table 2 biomedicines-13-02539-t002:** Summary of measurement of sleep quality and cardiovascular among healthcare shift workers of the included studies.

Author (Country)	Sample Size (*n*)	Exposure of Working Shift	Cardiovascular Measures	Sleep Quality Measures	Outcome	Conclusions
Lajoie P., 2015 (Canada) [31]	Shift female hospital employees (*n* = 121)	Rotating 12 h day/night shifts	Metabolic syndrome	PSQI -sleep latency -sleep efficiency	**CVS result:** Rotating 12 h shift work was associated with two fold increased risk of metabolic syndrome (OR = 2.29) among female hospital workers. **Sleep result:** Night shift workers had significant poor sleep quality than day workers with 47.9% scoring > 5 on the PSQI compared to 32.7% of day workers (*p* < 0.01). They also reported poor sleep latency (42% vs. 27%) and lower sleep efficiency, indicating reduced sleep quality.	Shift work strongly associated with metabolic syndrome among female workers however sleep quality was not a significant mediator and no significant association with metabolic syndrome.
Silva-Costa A, 2015 (Brazil) [32]	Night shift nurses (*n* = 340)	Work night shifts at least once a week or four times per month	Self-reported physician diagnosed cardiovascular disease: hypertension, coronary disease, myocardial infarction, or heart failure	Insomnia	**CVS result:** The proportion of self-reported physician diagnosed CVD was 18% among day workers and 21% among night workers. **Sleep result:** The prevalence among night workers reported that 24% had nocturnal insomnia, 35% had diurnal insomnia, 13% experienced both while 23% of day workers reported nocturnal insomnia.	Night workers who had insomnia during both sleep episodes had three times higher odds of developing cardiovascular disease compared to those without insomnia.
Hsu HC, 2021 (Taiwan) [16]	Female shift work nurses (*n* = 393)		Heart rate variability (HRV): TP, LF, HF, LF/HF, SDNN, RMSSD	PSQI	**CVS result:** Reduced autonomic function (lower TP) and higher mean heart rate are moderately predictive of poor sleep quality. Lower low frequency and total power and higher high frequency showed an altered autonomic balance. **Sleep result:** 95.9% had poor sleep quality, scoring 10.2. The majority had sleep efficiency less than 85%, and 2.8% reported no difficulty falling back asleep after waking.	Poor sleep quality correlated with lower TP and LF. Poor sleep in nurses is linked to autonomic imbalance, potentially elevating cardiovascular risk.
Panwar A, 2024 (India) [33]	Female nurses (*n* = 38)	Morning (9 a.m. to 2 p.m.) shift and night shift (9 p.m.–9 a.m.)	HRV: frequency and time domains	PSQI	**CVS result:** Decrease in standard deviation of normal-to-normal intervals, total power, and high-frequency power while increase in LF/HF ratio. **Sleep result:** Night shift nurses had significantly poorer sleep quality than morning shift nurses.	Night shift work is related to poor sleep quality and significant disruptions of heart rate variability.
Liao YH, 2025 (Taiwan) [34]	Shift-work nurses (Intervention group: 32; Control group: 32) (*n* = 64)	4-week shift work schedule in the last 6 months (day, evening, night shift) LLLT using red and near-infrared light, administered three times a week for one month	Heart rate variability	Insomnia Severity Index	**CVS result:** No significant difference between in HRV outcomes. **Sleep result:** Intervention group resulted 4.3, controls 12.6 (*p* < 0.001), reflecting significantly less severe insomnia in the LED-LLLT group.	LLLT was effective in improving in shift-work nurses with insomnia. However, it did not cause significant changes in heart rate variability.

Abbreviations: cardiovascular (CVS); cardiovascular disease (CVD); low frequency/high frequency (LF/HF); heart rate variability (HRV); total power (TP); low frequency (LF); high frequency (HF); standard deviation of NN intervals (SDNN); Root Mean Square of Successive Differences (RMSSD); low-level LED light therapy (LLLT).

**Table 3 biomedicines-13-02539-t003:** Summary of measurement of sleep quality, cortisol, and cardiovascular among healthcare shift workers of the included studies.

Author	Sample Size (*n*)	Exposure of Working Shift	Cortisol Measure	Cardiovascular Measure	Sleep Quality Measure	Outcome	Conclusions
Roy, 2023 (India) [35]	Healthcare providers: 60 fixed duty and 60 shift duty (*n* = 120)	Fixed time schedule: 10.00–17.00 (Monday to Friday) Shift duty schedule: Two morning (6:00–14:00), two evening (14:00–22:00), two night duties (22:00–6:00)	Blood	HRV (SDNN, RMSSD, NN50, pNN50, mean HR)	Athens Insomnia Scale	**Cortisol result:** Shift duty workers had significantly higher evening cortisol levels (9.4 ± 2.36 mcg/dL) than fixed duty workers (3.74 ± 1.7 mcg/dL) (*p* = 0.036). **Cardiovascular result:** Low HF power. LF/HF ratio significantly higher (*p* < 0.001) in shift workers **Sleep result:** Shift duty workers had a higher chance of having insomnia (Athen’s score > 6) (*p* < 0.001) than fixed duty workers.	Long duration of shift work increases evening cortisol level and increases the chance of having insomnia and impaired cardiovascular regulation.

Abbreviations: standard deviation of all normal-to-normal (SDNN) intervals; Root Mean Square of Successive Differences (RMSSD) intervals; number of successive NN pairs differing by more than 50 ms (NN50); percentage of NN intervals differing by greater than 50 ms (pNN50); Heart Rate (HR).

## Data Availability

Data can be obtained from the corresponding author upon reasonable request.

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
