# Peer review of "A Systematic Review of Literature on the Association Among Sleep, Cortisol Level and Cardiovascular Health Within the Healthcare Shift Worker Population"

_biomedicines, 2025, doi:10.3390/biomedicines13102539_

Round 1
Reviewer 1 Report
Comments and Suggestions for Authors
Manuscript ID: Biomedicines-3894294
Title: A Systematic Review of Sleep Quality, Cortisol Dysregulation, and Cardiovascular Health Among Healthcare Shift Workers
The authors of this review included relevant literature related to sleep quality, cortisol level and cardiovascular health in a population of healthcare shift workers published in SCOPUS, WoS, and PubMed databases from 2015 and 2025. They tried to focus on a possible association among the three groups of variables in the above population. The area of review is interesting. However, I found out several issues in the manuscript (outlined below) which should be addressed by the authors adequately.
Title: The title of the manuscript does not reflect the area of the review adequately. I suggest that the title should read like this: “A Systematic Review of Literature on the Association among Sleep, Cortisol Level and Cardiovascular Health within the Healthcare Shift Worker Population.” This is just a suggestion. The authors may revise it accordingly.
Line #61: What is ‘fixed shift workers?’ Do the authors mean permanent shift work?
Line #64: What is sleep activity? Normally there is the least activity during the sleep. Do the authors mean ‘sleep-wake rhythm?’
Line #85: What are endocrine parameters of cortisol? The authors may consider revising this.
2.5 Study quality: What are these abbreviations, such as ANAS, NSJ, and MI? For the sake of brevity and clarity for the general readers of this review, these abbreviations should be explained.
3.1 Studies selected: This section should have been appropriately placed in the 2. Methods section.
Line #143: What are the different parameters of cortisol? It is difficult to comprehend. The authors may please explain this.
Line #162: What is the threshold limit for the morning cortisol? The threshold limit is undefined.
Line #184: What are the reference values?
Lines #193-196: This statement is very much confusing. The authors may revise this statement to make it more comprehendible.
Line #202: What do the authors mean by high sleep quality? How to define it?
Line #208: What is day shift worker?
Lines #283-286: How does shift work induce circadian disruption? How do the authors define circadian disruption?
Line #346: What does ‘shift female hospital workers’ mean? The authors may revise this.
Lines #356-360: The authors may please revise this statement to make it more comprehensible.
The authors might have presented a cartoon diagram or a schematic diagram for depicting their conclusion.
Comments on the Quality of English LanguageThe quality of English Language is required to be improved.
Author Response
Please find the attached file for your kind reference.

Reviewer 2 Report
Comments and Suggestions for Authors
The article provides valuable insights into a relevant occupational health issue. However, several methodological and conceptual limitations undermine the strength of its conclusions. Authors can ‘easily’ improve the paper. My recommendations:
- Method: narrow database selection: The review relied exclusively on PubMed, Scopus, and Web of Science. While these are major sources, excluding PsycINFO, Embase, and CINAHL may have led to an incomplete retrieval of relevant literature, especially given the psychological and nursing-related focus of the topic. Just add this as limitation perceived by authors;
- search strategy: search terms, although covering core concepts (sleep, cortisol, cardiovascular outcomes, shift work), lack breadth. Synonyms and related concepts such as “stress hormones,” “autonomic function,” “occupational fatigue,” or “sleep deprivation” might have retrieved additional studies. If authors want to develop more, I recommend;
- clarify more why qualitative or mixed-methods research was excluded, despite their potential to capture psychosocial and contextual aspects of shift work;
- sample: only 14 studies were included, most of which involved nurses. This raises concerns about generalizability to other healthcare workers. Add more (in depth) justification;
-high heterogeneity: Cortisol was measured in different biological matrices (saliva, blood), at different times of day, and with inconsistent sampling protocols. Likewise, sleep was assessed with multiple tools (PSQI, AIS, ESS, actigraphy), complicating comparisons. Cardiovascular health outcomes ranged from HRV indices to self-reported diagnoses, adding further inconsistency;
- the authors mention using the JBI tool, but they do not provide detailed quality scores for each study;
- while the review acknowledges variability, the discussion sometimes suggests stronger causal links than the data support. For example, the cyclical relationship between poor sleep, cortisol dysregulation, and cardiovascular risk is plausible but not definitively demonstrated by the reviewed studies;
- many included studies did not adequately control for lifestyle variables (diet, physical activity, caffeine intake, psychosocial stress), which also affect cortisol, sleep, and cardiovascular outcomes. This limitation is underemphasized in the review;
- there are insufficiency regarding literature, for example Goldstein should be added as citation (regarding circadian rhythms);
- publication bias: no formal assessment of publication bias was reported. Positive findings linking shift work to health risks may be overrepresented;
- cortisol: the review positions cortisol as a central biomarker but neglects other key physiological mediators (e.g., melatonin, inflammatory cytokines, autonomic nervous system activity beyond HRV). This narrow focus restricts the scope of conclusions;
- although the paper briefly mentions light therapy, it excluded intervention trials from its main analysis. This limits practical recommendations for mitigating health risks in shift workers;
- redundancy and repetition: The discussion reiterates findings without always synthesizing them into new insights;
- some results are presented without clear effect sizes, confidence intervals, or strength of evidence. This weakens the reader’s ability to assess the robustness of associations;
- conclusion notes future research needs, it does not provide specific, actionable guidance for improving study designs.
Many thanks, please assure that you perform revisions as you find suitable.
Author Response
Please find the attached file for your kind reference

Round 2
Reviewer 1 Report
Comments and Suggestions for Authors
No further suggestions.
Author Response
Comment 1: No further suggestions.
Response 1: We thank the reviewer for their time and constructive feedback. We appreciate that no further suggestions were made.